# Flower consumption, ambient temperature and rainfall modulate drinking behavior in a folivorous-frugivorous arboreal mammal

Óscar M. Chaves[1,2]*, Vanessa B. Fortes[3], Gabriela P. Hass[1], Renata B. Azevedo[4], Kathryn E. Stoner[5], Júlio César Bicca-Marques[1]

**1** Escola de Ciências da Saúde e da Vida, Pontifícia Universidade Católica do Rio Grande do Sul, Porto Alegre, Rio Grande do Sul, Brazil, **2** Escuela de Biología, Universidad de Costa Rica, San José, Costa Rica, **3** Departamento de Zootecnia e Ciências Biológicas, Universidade Federal de Santa Maria, Rio Grande do Sul, Brazil, **4** Centro Nacional de Pesquisa e Conservação de Primatas Brasileiros, Instituto Chico Mendes de Conservação da Biodiversidade, João Pessoa, Paraíba, Brazil, **5** Department of Fish, Wildlife and Conservation Biology, Colorado State University, Fort Collins, Colorado, United States of America

* ochaba@gmail.com

**Data Availability Statement:** All relevant data needed to replicate the analyses are within the paper and its Supporting information files.

## Abstract

Water is vital for the survival of any species because of its key role in most physiological processes. However, little is known about the non-food-related water sources exploited by arboreal mammals, the seasonality of their drinking behavior and its potential drivers, including diet composition, temperature, and rainfall. We investigated this subject in 14 wild groups of brown howler monkeys (*Alouatta guariba clamitans*) inhabiting small, medium, and large Atlantic Forest fragments in southern Brazil. We found a wide variation in the mean rate of drinking among groups (range = 0–16 records/day). Streams (44% of 1,258 records) and treeholes (26%) were the major types of water sources, followed by bromeliads in the canopy (16%), pools (11%), and rivers (3%). The type of source influenced whether howlers used a hand to access the water or not. Drinking tended to be evenly distributed throughout the year, except for a slightly lower number of records in the spring than in the other seasons, but it was unevenly distributed during the day. It increased in the afternoon in all groups, particularly during temperature peaks around 15:00 and 17:00. We found via generalized linear mixed modelling that the daily frequency of drinking was mainly influenced negatively by flower consumption and positively by weekly rainfall and ambient temperature, whereas fragment size and the consumption of fruit and leaves played negligible roles. Overall, we confirm the importance of preformed water in flowers to satisfy the howler's water needs, whereas the influence of the climatic variables is compatible with the 'thermoregulation/dehydration-avoiding hypothesis'. In sum, we found that irrespective of habitat characteristics, brown howlers seem to seek a positive water balance by complementing the water present in the diet with drinking water, even when it is associated with a high predation risk in terrestrial sources.

Additional datasets used to perform the main statistical analysis are available in Mendeley Data (http://dx.doi.org/10.17632/3gxy6vrsbf.1) and Dryad Digital Repository (https://doi.org/10.5061/dryad.hdr7sqvh7).

**Funding:** This study was supported by a grant from the Programa Nacional de Pós-Doutorado of the Brazilian Higher Education Authority/CAPES (PNPD grant # 2755/2010). OMC was supported by a PNPD postdoctoral fellowship. JCBM thanks the Brazilian National Research Council/CNPq for research fellowships (PQ#303306/2013-0 and 304475/2018-1). GPH was supported by a doctoral fellowship from CNPq (GD#140641/2016-5). The funders had no role in study design, data collection and analysis, decision to publish, or preparation of the manuscript.

**Competing interests:** The authors have declared that no competing interests exist.

# Introduction

Water is an essential chemical substance for all animals, not only because it represents a large percentage of whole-body mass, but because it is the medium within which the chemical reactions and physiological processes of the body take place [1–3]. This substance is involved in a myriad of vital processes, such as secretion, absorption, and transport of macromolecules (e.g. nutrients, hormones, metabolites, antibodies, and neurotransmitters), electrolyte homeostasis, transmission of light and sound, and thermoregulation [2–4]. Therefore, water intake is essential for animal health and survival, particularly in the case of terrestrial vertebrates [3, 5–7].

In all terrestrial mammals, water inputs come from three major sources—water ingested within consumed foods, metabolic water resulting from macronutrient oxidation, and water drunk. Water outputs result from excretion, egestion, or evaporation through the skin or the respiratory tract [4, 5, 8]. When water intake is appropriate, healthy animals maintain a physiological state in which water inputs and outputs are the same throughout the day (i.e. the 'water balance'), an essential condition for the correct functioning of body cells [2, 5]. Animals reach this water and electrolyte homeostasis by applying a repertoire of behavioral and physiological strategies that depend on the organism's complexity and the surrounding environment [2]. Whereas drinking increases water input, shade seeking, low metabolic rates, and the excretion of salt by the kidney reduce water loss [1, 2, 4]. Dehydration (i.e. a negative water balance) resulting from long periods of adverse dry conditions when water losses exceed water intake can seriously compromise health, being lethal when losses reach 15 to 25% of body weight (camels are an exception [2, 4]).

Given that plant items contain more water than animal bodies, herbivorous mammals are expected to obtain a larger volume of water from their diets than do omnivores and carnivores [8]. However, plant items can show wide intraspecific and seasonal variations in chemical composition that influence their importance and reliability as water sources, thereby influencing the animals' need to drink [9]. Herbivorous mammals inhabiting dry environments, such as desert rodents and camelids, can reach water balance by relying on preformed (i.e. water in plant items) and oxidation (i.e. metabolic water resulting from macronutrient oxidation) water during dry periods [2, 10]. In addition to these water sources, animals inhabiting wetter environments also rely on another major source, drinking water [2, 7, 11]. Drinking is rare (e.g. giraffe, *Giraffa camelopardalis*) or presumably nonexistent in mammals that rely on succulent diets [2]. Arboreal folivores once believed to obtain all their water demands from food have been reported to drink either in captivity (sloth, *Choloepus hoffmanii* [12]) or in the wild (koala, *Phascolarctos cinereus* [13, 14]).

While ground-living species drink water from rarely-depletable sources (e.g. rivers, streams, and lagoons), highly arboreal mammals depend on depletable arboreal reservoirs, such as bromeliads and treeholes (primates [15–18]), or on short lasting rain water on tree branches and leaves (koalas [14], sloths [19]). The exploitation of terrestrial water reservoirs by the latter tends to be rare because their vulnerability to predators likely increases when they descend to the forest floor, as has been observed for tropical primates [15, 20–23].

Among the highly arboreal Neotropical primates, reports of drinking are restricted to a few social groups of the better-studied taxa, including howler monkeys (*Alouatta* spp. [6, 15, 17, 22, 24–26], spider monkeys (*Ateles geoffroyi* [27]), capuchin monkeys (*Cebus capucinus* [28], *Sapajus libidinosus* [29]), and marmosets (*Callithrix flaviceps* [11]). These monkeys meet their water needs primarily via preformed water [15, 30], although they also drink from arboreal reservoirs or, to a lesser extent, terrestrial sources [15–17, 20].

Two main non-exclusive hypotheses have been proposed to explain the drinking behavior of howler monkeys. The thermoregulatory/dehydration-avoidance hypothesis (TDH) relates

drinking to a behavioral strategy for maintaining a positive water balance during the hottest and driest periods of the year [6, 17, 26]. The metabolite detoxification hypothesis (MDH) states that the consumption of large amounts of some plant parts (e.g. mature leaves, branches, and seeds) containing digestion inhibitors (fiber and secondary metabolites) 'forces' monkeys to drink more to help in their processing [15, 17, 20, 26]. The trend of anti-herbivory metabolites to increase in plants with increasing latitude [31] further supports the potential relevance of the MDH to howler monkeys living in southern latitudes (e.g. *Alouatta guariba clamitans* and *A. caraya*). The bacterial fermentation of the leaf-rich diet of howlers also requires an appropriate water supply [32], as does the excretion of the higher salt content of leaves [8].

Howlers' low rates of digestion [32] together with the cumulative water loss via urine, lung evaporation, and sweat over the course of an activity period (i.e. daytime), especially during more active and hot times, and under low air humidity [33], can increase plasma osmolarity and cell dehydration to levels that cause thirst and create circadian rhythms of drinking [2, 30]. Similar drinking rhythms associated with feeding are found in squirrel monkeys (*Saimiri* sp. [34]) and owl monkeys (*Aotus* sp. [35]). Finally, forests inhabited by howler monkeys also show seasonal and site-related differences in thermal environment [36], food availability [6, 37], and the presence and reliability of water sources. Therefore, it is important to identify the factors that modulate their drinking behavior to better understand how habitat patch size and spatial restriction resulting from land use changes can affect their health and survival.

Here we investigate the drinking behavior in wild groups of brown howler monkeys (*A. guariba clamitans*) inhabiting Atlantic Forest fragments in southern Brazil as models of folivorous-frugivorous arboreal mammals. Specifically, we assess (i) the arboreal and terrestrial water sources that these monkeys exploit and how they drink, (ii) the daily frequency and seasonal distribution of drinking records, and (iii) the influence of fragment size, season, ambient temperature, rainfall, and the contribution of fruits, leaves, and flowers to the diet on drinking. We predicted that brown howlers would complement the preformed water obtained from their diet with water from arboreal and terrestrial reservoirs, if available, because the availability of fleshy fruits, flowers and young leaves vary seasonally in the study region [37]. We also predicted a within-day gradual increase in drinking in the afternoon in response to an increase in water demands resulting from temperature rise throughout the day and the daily water loss via digestion, excretion, breathing, and sweating [1, 8]. Finally, we predicted that diet composition, climatic variables, and fragment size influence the frequency of drinking. While the TDH will receive support if ambient temperature and rainfall are good predictors of the frequency of drinking, a positive influence of leaf ingestion on water consumption will support the MDH.

## Methods

This investigation followed the ethical guidelines of the International Primatological Society and the legal requirements established by the Ethical Committee of the Zoological Society of London for research with nonhuman primates. All studies met all Brazilian animal care policies and were strictly observational. Furthermore, studies conducted from 2011 to 2019 were approved by the Scientific Committee of the Faculty of Biosciences of the Pontifical Catholic University of Rio Grande do Sul (projects SIPESQ #5933 and 7843).

### Study fragments and groups

We collected data on 14 groups of wild brown howlers inhabiting Atlantic Forest fragments ranging from 1 to 977 ha in the municipalities of Porto Alegre, Viamão, and Santa Maria in the State of Rio Grande do Sul, southern Brazil (Table 1, Fig 1). We classified the fragments in

**Table 1. Study fragments, brown howler group size, sampling effort, and number of drinking records.**

| Site | Size | Latitude | Longitude | WS[a] | Group size[b] | Sampling effort | | | #rec.[d] |
|---|---|---|---|---|---|---|---|---|---|
| | | | | | | Months | Days[c] | Hours | |
| S1 | 1.6 | S30˚11' 00.1" | W51˚06' 06.6" | P,B | 7 (1,2) | 21 | 67 (4) | 492 | 4 |
| S2 | 9.5 | S30˚12' 18.4" | W51˚06' 05.7" | R,B | 11 (1,3) | 19 | 61 (13) | 438 | 47 |
| S3 | 2.3 | S30˚12' 26.6" | W51˚05' 54.0" | S,P,B | 10 (1,3) | 20 | 65 (12) | 539 | 28 |
| S4 | 3.6 | S30˚12' 27.0" | W50˚55'39.0" | B,T | 8 (1,2) | 12 | 56 (33) | 681 | 90 |
| S5 | 5.2 | S30˚17'27.0" | W50˚57'36.0" | B,T | 4 (1,2) | 12 | 55 (14) | 663 | 27 |
| S6 | 7.3 | S30˚17'39.0" | W51˚00'42.0" | B,T | 8 (1,2) | 12 | 69 (9) | 826 | 13 |
| S7 | 1 | S29˚47'05.8" | W53˚53'12.0" | S,B | 7 (1,4) | 12 | 59 (44) | 654 | 322 |
| M1 | 27 | S30˚12' 00.0" | W51˚04'00.0" | P,B | 12 (2,2) | 12 | 57 (34) | 518 | 173 |
| M2 | 17 | S29˚45'21.3" | W53˚52'32.2" | S,B | 6 (1,2) | 12 | 58 (26) | 623 | 99 |
| L1 | 108 | S30˚10' 39.5" | W51˚06' 18.2" | R,B,S | 9 (2,3) | 21 | 73 (6) | 536 | 18 |
| L2 | 93 | S30˚23' 15.6" | W51˚02' 43.3" | L,P,B | 12 (3,3) | 18 | 81 (7) | 460 | 10 |
| L3 | 106 | S30˚20' 56.8" | W51˚02' 58.2" | L,P,B | 10 (2,3) | 17 | 87 (27) | 536 | 102 |
| L4 | 977 | S29˚46'46.0" | W53˚51'52.0" | S,B | 5 (2,3) | 12 | 54 (45) | 577 | 184 |
| L5 | 977 | S29˚47'05.9" | W53˚53'03.0" | S,B | 7 (2,3) | 12 | 67 (39) | 836 | 144 |
| **Sum** | | | | | **116** | **212** | **909 (313)** | **8379** | **1261** |

[a] Water sources detected during the study period: Bromeliads (B), treeholes (T), pools (P), rivers (R), streams (S), and lagoon (L).

[b] Group size and number of adult males and females (in parentheses).

[c] Number of study days with drinking events in parentheses.

[d] Total number of drinking records per study group.

three size categories: small (<1 to 10 ha), medium (>10 to 100 ha) and large (>100 to 1,000 ha; *sensu* [38]). Small and medium fragments in Porto Alegre (S1-S3 and M1) and Viamão (S4-S6; Fig 1) were surrounded by anthropogenic matrices comprised of small human settlements, pastures, subsistence orchards, and small parcels of cultivated land (<0.5 to 2 ha). None of them are officially protected. Conversely, the large fragments in Porto Alegre and Viamão (L1-L3) are found in legally protected areas (Fig 1, see [37] for further information on these fragments). The Atlantic Forest study fragments in Santa Maria (S7, M2, L4, and L5; <1 to 977 ha; Fig 1) compose a 5,876-ha mosaic of natural grasslands, extensive pastures devoted to cattle ranching, and other scattered forest fragments. This area, named Campo de Instrução de Santa Maria (CISM), belongs to the Brazilian Army. Therefore, although it is not officially protected by Brazilian laws, CISM is impacted by a lower human pressure than the unprotected study sites.

The predominant vegetation in all study fragments is subtropical semideciduous forest. Despite differences in size, all fragments have a similar vegetation structure with a canopy often <30-m tall and a relatively open understory allowing clear observations of monkeys in the canopy. Given its latitude (30˚-31˚S), the region is characterized by marked climatic seasonality: summer (21 December-20 March), fall (20 March-21 June), winter (21 June-22 September), and spring (22 September-21 December). According to meteorological records of Porto Alegre, the average annual ambient temperature during the study period was 21˚C [39]. The highest temperatures occurred in the summer (mean = 26˚C, range = 19˚-35˚C), and the lowest in the winter (mean = 15˚C, range = 3˚-26˚C; S1 and S2 Figs in S1 File). The average annual rainfall during the study years was 1,450 mm. There was no clear rainfall pattern between months or seasons in Porto Alegre, Viamão or Santa Maria (S1 and S2 Figs in S1 File).

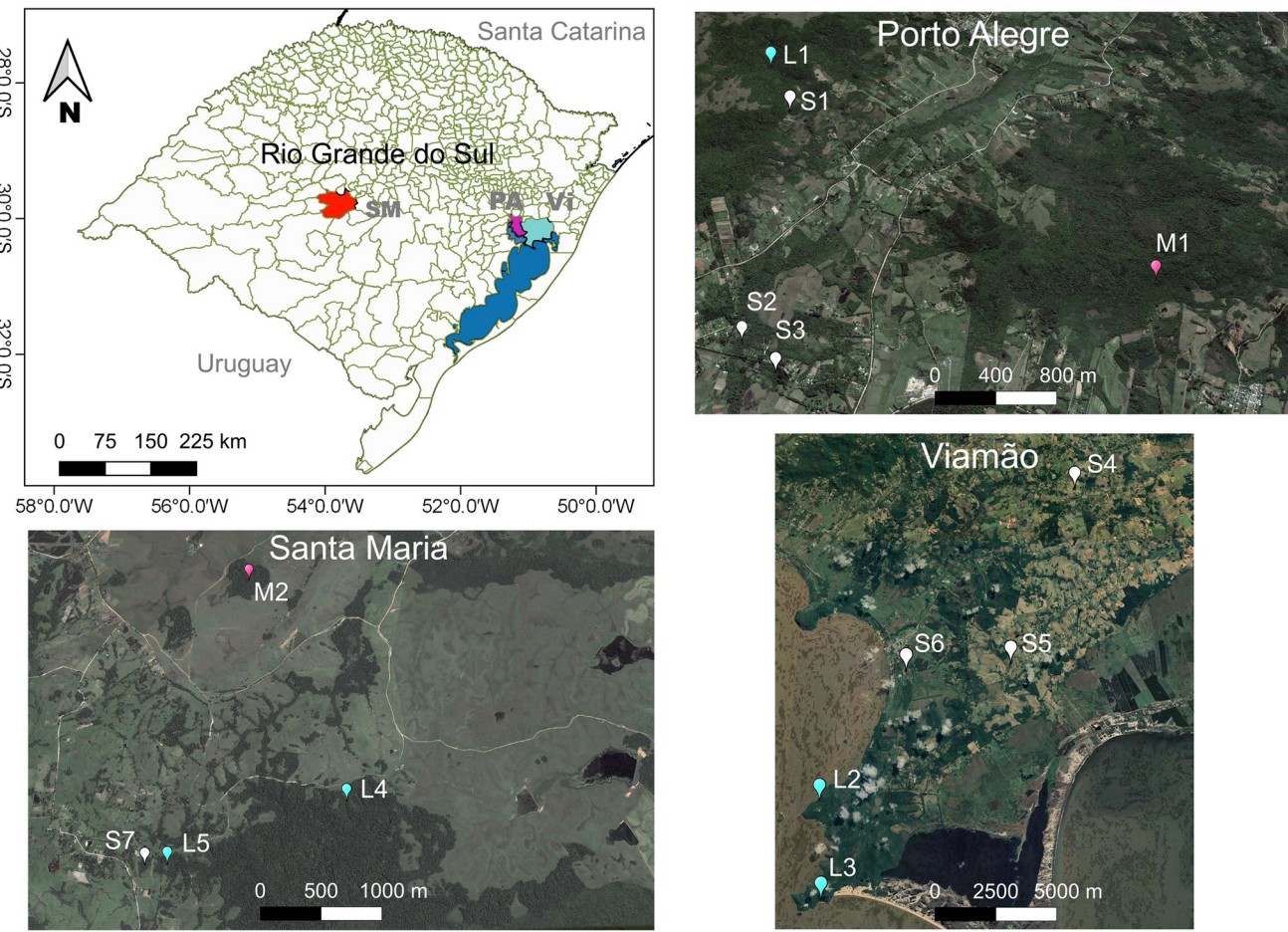

**Fig 1. Location of the 14 study sites in the municipalities of Santa Maria (SM, red polygon), Porto Alegre (PA, purple polygon), and Viamão (Vi, cyan polygon), southern Brazil.** Color markers indicate the exact location of the small (white), medium (rose), and large (cyan) Atlantic forest fragments inhabited by the study groups. Lansat7 open-access images (available at http://earthexplorer.usgs.gov/) from 2008 for SM and 2013 for PA and Vi.

Despite the variation in fragment size, all fragments contained fleshy fruit tree species (i.e. *Ficus* spp., *Eugenia* spp., and *Syagrus romanzoffiana*) intensively exploited by brown howlers [37, 40, 41]. All study fragments had arboreal and terrestrial water reservoirs, such as bromeliads, streams, and/or rivers (Table 1).

We followed brown howler groups ranging from 4 to 12 individuals in each fragment (*n* = 116 individuals, Table 1). All individuals of the study groups in small fragments were well-habituated to humans before the study, while we habituated the individuals of the study groups inhabiting medium and large fragments during two to three months prior to their respective monitoring. We followed Williamson & Feistner's [42] recommendations for the habituation of free-ranging primates. Whenever possible, we kept a distance of 15 to 30 m from the projected position of the animals on the ground, and we wore green or olive clothes. Additionally, the same observer (OMC, VBF, GPH, or RBA) habituated and followed the study group(s) during each study period in each fragment. Whereas most groups inhabited a single forest fragment, S1, S2, and S7 (hereafter named by the acronym of their respective fragments) used more fragments. S1 ranged outside of its most used fragment for about 35% of the study days to feed in a neighboring 10-ha fragment. S2 regularly used three forest fragments distant about 70 to 850 m from each other (the home range of this group included the area of these three fragments). Lastly, S7 also used three forest remnants distant from 30 to 40 m from each other.

## Behavioral data collection

We studied the diet and drinking behavior of the groups during periods ranging from 12 to 21 months (Table 1): (i) January to December 1996 (L5), (ii) June 2002 to August 2003 (M1), (iii) January to December 2005 (S7, M2, and L4), (iv) June 2011 to June 2014 (S1, S2, S3, L1, L2, and L3), and (v) June 2018 to July 2019 (S4, S5, and S6). We collected data for all groups from dawn to dusk using high-resolution 10 x 42 binoculars. We monitored the groups on a monthly basis during three to eight consecutive days in periods (i), (ii), and (iii), during four to five consecutive days on a bimonthly basis in period (iv), and during four to nine consecutive days on a monthly basis in period (v). We recorded the behavior (including feeding and drinking) of these groups using the instantaneous scan sampling method in periods (i) to (iv) and the 'all occurrences' method [43] to record all drinking events (i.e. when one or more members of the study group drank) that occurred outside scan sampling units. We collected 5-min scan samples at 15-min intervals. Finally, we used the focal-animal method [43] to record the behavior (including feeding and drinking) and complemented the recording of all events of drinking by any group member also using the 'all occurrences' method. We recorded the behavior of the focal howler instantaneously at 20-s intervals during 5-min focal sampling units every 15 min. A single focal adult was monitored per group during each study period in order to fulfill the primary goal of the respective research.

We recorded the feeding and drinking behavior of adults, subadults, and conspicuous juvenile individuals, except for S4, S5, and S6, of which we only recorded the behavior of adults. We assumed that our diet composition datasets based on two sampling methods are comparable because the within-group feeding behavior of howlers is very similar among age-sex classes [44–46].

During feeding bouts we recorded the main plant items eaten (i.e. ripe and unripe fruits, adult and young leaves, and flowers) and, whenever possible, the plant species (see [37] for additional details). When a howler drank water, we recorded the individual's identity, the type of arboreal or terrestrial water source (i.e. bromeliad, treehole, river, stream, or pool), and the water acquisition method (e.g. inserting the head in the treehole and drinking the water directly with the mouth or inserting a hand in the water and licking the dripping water). Whenever possible, we also identified the bromeliad source of water at the species level. We used the number of drinking records (i.e. the total number of records devoted to drinking per study day) and the number of feeding records devoted to each plant item in the analyses.

## Climatic data

We obtained data on ambient temperature and rainfall for the three study municipalities from the meteorological database of the Instituto Nacional de Meteorologia do Brasil [39]. For each day with a record of drinking, we used the meteorological data for ambient temperature for that day and the weekly rainfall (i.e. the rainfall accumulated during the previous seven days) as proxies for the thermal environment and the amount of rainfall water potentially available for brown howlers. Furthermore, we recorded the hourly variation in in-site ambient temperature throughout the day during periods (iv) and (v) to determine daily peaks. We measured the temperature after each scan or focal sampling unit in the shade at a height of ca. 2 m above the ground using a pocket thermo-hygrometer (Yi Chun®, PTH 338) during period (iv) and a portable meteorological station (Nexus, model 351075) distant from 0.4 to 2.6 km from the study fragments during period (v). We used the data on ambient temperature as a predictor variable in our modelling and the hourly variations in temperature to prepare S5 Fig in S1 File.

## Statistical analyses

We performed Chi-square tests for proportions at the group level to compare the proportions of drinking records per water source and season in each study group using the 'prop.test' function of R. We calculated these proportions by dividing the number of records for each water source or season (i.e. summer, fall, winter, spring) by the total number of records for each group during the entire study period. We did not perform between-fragment comparisons because of sampling effort differences (i.e. the number of sampling months, days, or days per month varied between the five study periods, Table 1). We used the same procedure above to calculate and compare the proportion of drinking records in each hour of the day in those fragments with >10 drinking records. When we found significant differences, we compared the proportion of records in each class using post-hoc proportion contrasts via the R function 'pairwise.prop.test' with a Bonferroni correction because of multiple comparisons of the same data sets.

We performed a generalized linear mixed-effects model (GLMM) to assess the influence of six predictor variables—contribution of fruits, leaves, and flowers to the diet, fragment size (i.e. fragment area in hectares), ambient temperature, and weekly rainfall—on the total number of drinking records recorded from dawn to dusk using the function 'lmer' of the R package lme4. Although the number of drinking records was not correlated with group size ($r_s = 0.03$, $P = 0.53$), we used the latter as a covariable in the model to control for any potential influence of group size differences (Table 1) on the response variable. Similarly, we used 'sampling method' (instantaneous scan and focal animal) as a covariable. We set the Poisson error family for the response variable as recommended for count data [47, 48]. We performed the overdispersion diagnostic via the R package DHARMa. Even when data are overdispersed (overdispersion ratio $\omega = 1.4$, $P = 0.03$), statistical adjustments (e.g. set a negative binomial distribution) are indispensable mainly when $\omega \geq 2$ [48]. We specified group ID as a random factor to account for repeated-measures from the same groups. We did not consider interactions between predictor variables to minimize overparameterization and problems of convergence of the global model (i.e. the model containing all fixed and random factors [49]). We standardized variable scales using the 'stdize' function of the R package MuMIn [50]. Additionally, we found no multicollinearity problem between variables using the 'vifstep' function of R package dplyr [51], as all of them had Variance Inflation Factor (VIF) <3 [52]. The correlation matrix between diet variables ($r_s$ ranged from 0.01 to -0.21) is available in S3 Fig in S1 File. Therefore, we included all variables in the global GLMM model.

We used the Akaike's Information Criterion for small samples (AICc) to select the models that best explain the effects of the predictor variables on drinking behavior. According to this criterion, the model with the strongest empirical support is the one with the smallest difference in AICc [53]. However, given that all models with ΔAICc<2 are considered equally parsimonious, we used the full-model averaging framework to determine which parameters best predict the number of drinking records while accounting for model uncertainty [49]. We used the 'dredge' function of the package MuMIn [50] to generate a full submodel set from the global model and the 'model.avg' function of the same package to determine the averaged model and the relative importance of each variable or predictor weight ($\Sigma w_i$). We used a likelihood ratio test over the function 'anova' to test the significance of the averaged model compared with the model including only the random factor (i.e. null model). We used the 'r.squaredGLMM' function of the package MuMIn to estimate an equivalent of the coefficient of determination or pseudo-$R^2$ for each competing best GLMM model. All statistical analyses were run in R v.3.6.3 [54] and the statistical significance threshold was set at $P \leq 0.05$.

## Results

### Water sources

We obtained a total of 1,261 individual drinking records (range = 4–322 records/group, Table 1) distributed in 917 drinking events and 313 observation days (range = 0–16 records/day, Table 1). We did not record drinking in 66% of the study days (i.e. 596 out of 909 days). The water sources were streams (44% of 1,258 records), followed by treeholes (26%), *Vriesea*, *Aechmea*, and *Tillandsia* bromeliads (16%), pools (11%), and rivers (3%) (Fig 2a). The

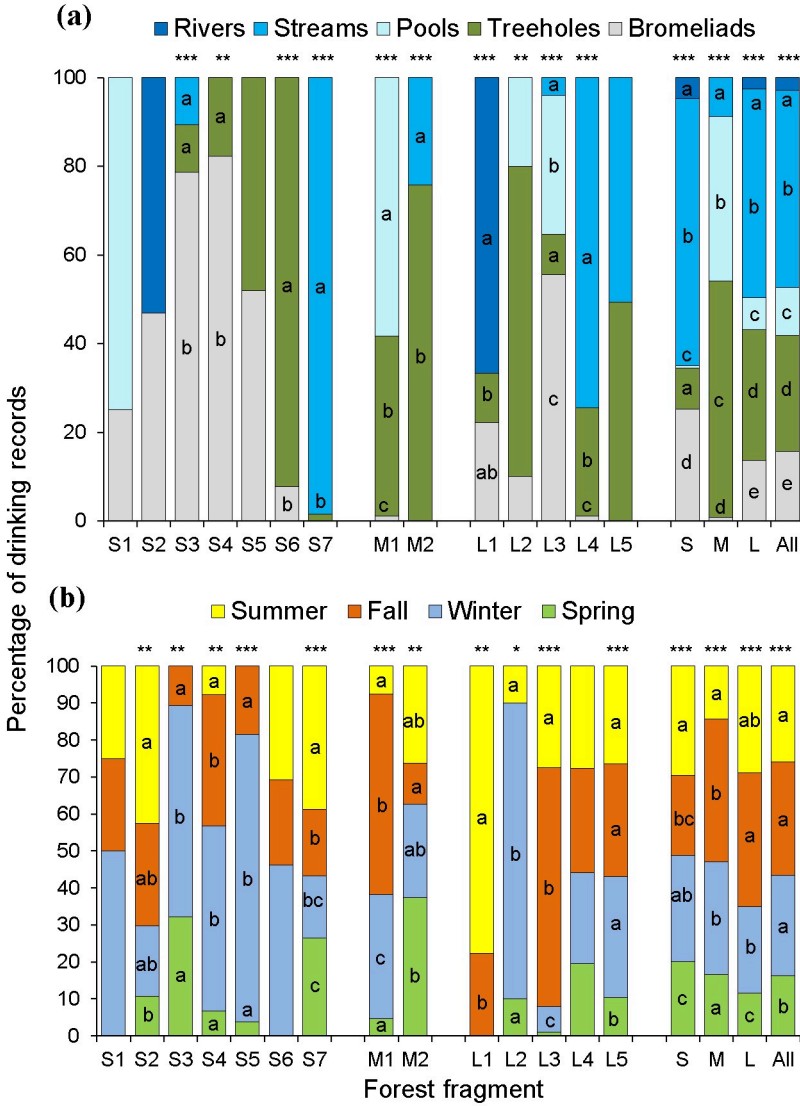

**Fig 2. Percentage of drinking records in 14 brown howler groups per water source (a) and season (b).** Different lower-case letters in the middle of each bar indicate significant differences in the number of records. Asterisks on the bars indicate the significance level according to Chi-square tests for proportions: $^{*}P \leq 0.05$, $^{**}P < 0.01$, $^{***}P < 0.001$. The proportion of records at the forest size level (small = S, medium = M, and large = L) and in the pooled dataset (All) is indicated in the four bars to the right. Water sources—rivers: Permanent water currents >4-m in width and >1-m in depth; streams: Seasonal water currents <2-m in width and <1-m in depth; treeholes: 10-40-cm diameter holes in trunks or large branches; bromeliads: Water stored in the rosette of epiphytic bromeliads. Significant differences in the proportion of records between water sources or seasons within each fragment are indicated with different lower-case letters in the bars. When proportion contrasts tests did not detect differences, no letter is shown. $N = 1,128$ records in (Fig 2a) and 1,131 records in (Fig 2b).

proportion of drinking records per water source type differed in nine of the 14 groups ($X^2$ tests, $P<0.05$ in all significant cases, Fig 2a). Arboreal sources were exploited by most groups (treeholes = 12, bromeliads = 11), whereas terrestrial ones were less common (streams = 6, pools = 4, rivers = 2; Fig 2a).

The most common drinking behavior consisted of inserting their head and sipping water directly from bromeliads and treeholes. When the treehole had a small diameter, monkeys immersed a cupped hand into the hole, pulled it out, and placed the mouth under the fingers to lick the dripping water. Vigilance was negligible during these arboreal drinking events.

In contrast, when drinking from terrestrial sources (rivers, streams and pools) howlers scanned the surroundings very carefully and were highly vigilant when drinking. Terrestrial drinking events began with some group members moving slowly to the understory, where they remained vigilant for ca. 30 s to 5 min before one or two of them descended to the ground to drink directly from the terrestrial water source for 102 ± 66 s (mean ± S.D., $n$ = 463) while the other individuals waited in vigilance in the understory. When the first individuals climbed back to the understory, the others descended slowly to the ground to drink, and the first remained in vigilance. Overall, a drinking event involved between one-fifth and four-fifths of the group members.

## Seasonal and daily patterns in drinking behavior

We found no clear pattern in the proportion of drinking records documented between or within seasons (Fig 2b, S4 Fig in S1 File). We observed drinking in all seasons in seven fragments, in three seasons in six fragments, and in two seasons in the remaining fragment (Fig 2b). We found seasonal differences in the proportion of drinking records in 11 out of 14 groups (proportion contrasts, $P<0.05$ in all significant cases, Fig 2b). A greater proportion of records from these groups occurred in a single season (winter—$n$ = 3 fragments: S3, S5, and L2; summer—$n$ = 2 fragments: S7 and L1; fall—$n$ = 2 fragments: M1, L3), in two seasons ($n$ = 1 fragment: S4) or in three ($n$ = 3 fragments: S2, M2, and L5; Fig 2b). We found a lower percentage of drinking records in the spring than in the other seasons when analyzing all groups together ($X^2$ = 77, df = 3, $P<0.0001$; Fig 2b).

Finally, we found that the distribution of drinking during the day showed a unimodal pattern in most fragments. The higher percentages of records occurred in the afternoon, particularly from 15:00 to 17:00 (8 out of 12 analyzed fragments; proportion contrasts, $P<0.05$ in all significant cases, Fig 3). This peak of drinking occurred near times with higher ambient temperatures in the fragments for which we have in-site temperature data ($n$ = 7; S5 Fig in S1 File).

## Factors driving the drinking behavior of brown howlers

We found nine models with substantial empirical support (i.e. ΔAICc<2). Together, these models included all predictor variables, except group size (Table 2). Flower consumption was the only predictor present in all models. The 'best model' for explaining the frequency of drinking also included ambient temperature, weekly rainfall, fruit consumption and sampling method, while the second 'best model' included the same variables, except fruit consumption and sampling method (Table 2).

The averaged model differed from the null model (likelihood ratio test: $X^2$ = 17, df = 6, $P<0.01$) and explained 40% of the variation in the number of drinking records. The three variables with higher predictive power in this model were flower consumption (β = -0.19, $\sum w_i$ = 0.97), ambient temperature (β = 0.12, $\sum w_i$ = 0.86), and weekly rainfall (β = -0.09, $\sum w_i$ = 0.79; Table 2). Finally, the focal-animal behavioral sampling method showed a negative influence on

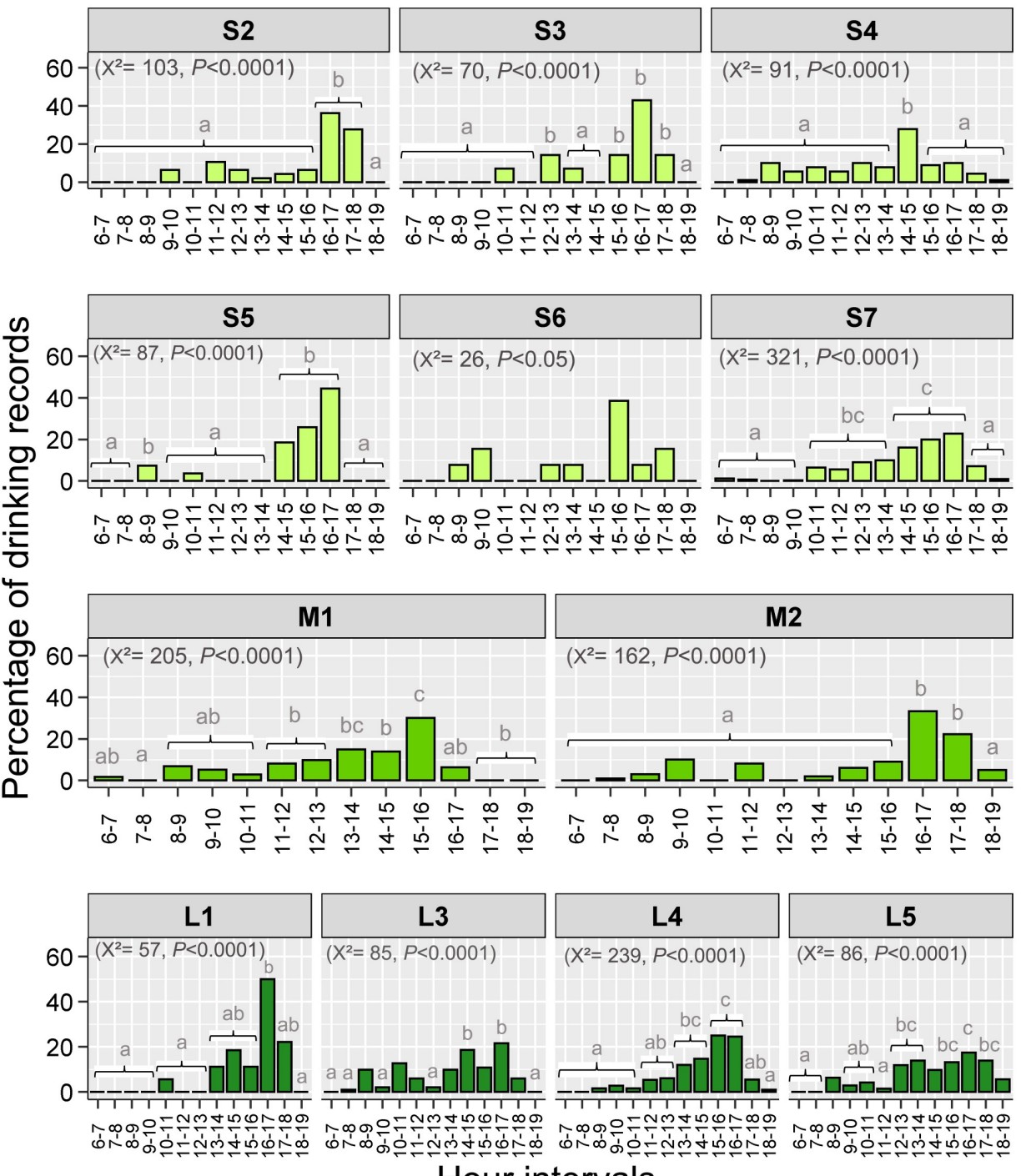

**Fig 3. Variation in percentage of drinking records by brown howler monkeys during the day in small, medium, and large Atlantic Forest fragments.** Different lower-case letters on the bars indicate significant differences after Bonferroni adjustment in *P* values. The absence of letters indicates that these hour intervals did not differ from the others. The number of degrees of freedom was 12 in all cases. Only fragments with >10 drinking records were considered in this analysis.

**Table 2. Best supported GLMM models (ΔAICc<2) and model-averaged that predict the variation in the number of drinking records in 14 brown howler groups in southern Brazil.**

| Predictor variables | Parameters[a] | | | |
|---|---|---|---|---|
| *Best supported models* | | | | |
| | $AIC_c$ | $\Delta AIC_c$ | $w_i$ | $R^2_c$ |
| 1) Flower+Fruit+Rainfall+Temp+Method | 1454.30 | 0.00 | 0.18 | 0.38 |
| 2) Flower+Rainfall+Temp | 1454.34 | 0.04 | 0.18 | 0.40 |
| 3) Flower+Temp+Method | 1455.18 | 0.89 | 0.12 | 0.39 |
| 4) Flower+Leaf+Rainfall+Temp +Method | 1455.31 | 1.01 | 0.11 | 0.40 |
| 5) Flower+Fruit+Temp+Method | 1455.50 | 1.21 | 0.10 | 0.37 |
| 6) Flower+Rainfall+Method | 1455.74 | 1.44 | 0.09 | 0.40 |
| 7) Flower+Fruit+Leaf+Rainfall+Temp+Method | 1455.91 | 1.61 | 0.08 | 0.38 |
| 8) Flower+Rainfall+Temp | 1455.94 | 1.64 | 0.08 | 0.43 |
| 9) Flower+Rainfall+Temp+Fsize | 1456.28 | 1.99 | 0.07 | 0.41 |
| *Averaged model ($R^2_c$ = 0.40)* | | | | |
| | $\beta_i$ | SE | 95% CI | $\Sigma w_i$ |
| Flower consumption (**Flower**) | -0.19 | 0.07 | (-0.32, -0.05) | **0.98** |
| Ambient temperature (**Temp**) | 0.12 | 0.07 | (0.00, 0.25) | **0.86** |
| Weekly rainfall (**Rainfall**) | -0.09 | 0.07 | (-0.24, 0.01) | **0.79** |
| Feeding behavior sampling method (Method) | | | | |
| Focal sampling | -0.54 | 0.30 | (-1.11, 0.06) | 0.74 |
| Fruit consumption (Fruit) | -0.04 | 0.08 | (-0.29, 0.05) | 0.37 |
| Leaf consumption (Leaf) | 0.02 | 0.05 | (-0.09, 0.25) | 0.28 |
| Fragment size (Fsize) | 0.01 | 0.08 | (-0.44, 0.65) | 0.18 |

[a]Akaike's Information Criterion for small samples (AICc), difference in AICc (ΔAICc), model probability Akaike weights ($w_i$), Pseudo-$R^2$ ($R^2_c$) indicating the percent of variance explained by the fixed and random factors, partial regression coefficients of the model-averaged ($\beta$), standard errors which incorporate model uncertainty (SE), 95% confidence intervals for the parameter estimates (95% CI), and relative importance of each predictor variable ($\Sigma w_i$) based in models with ΔAICc<4. The three predictor variables with higher contribution to the averaged model are enhanced in bold.

the number of drinking records, while the predicting importance of the other variables was minor (Table 2).

## Discussion

We found that brown howlers drank water accumulated in bromeliads and treeholes in the canopy, and that they also descended to the ground to drink from streams, rivers, and pools. Drinking increased in the afternoon and was less frequent in the spring. Also, while flower consumption and climatic conditions were good predictors of drinking behavior, the relevance of fruit and leaf consumption and habitat size was negligible.

The exploitation of non-food arboreal and terrestrial water reservoirs supports our expectation that oxidation and preformed water are insufficient for permanently satisfying howlers' water needs, as reported for many terrestrial mammals [1–3]. The finding that streams were the most used water sources by brown howlers differs from the greater importance of arboreal water reservoirs for other howler monkeys inhabiting both large (e.g. 1,564 ha in Barro Colorado Island, Panama [15, 30]) and small forest remnants (e.g. ≤10 ha [26, 55]).

This use of terrestrial water sources occurred despite the high risk of predation by domestic/stray dogs and small wild felids in the study region (e.g. *Leopardus wiedii* [56]; also OMC, personal observation). The fact that dog attacks represent a major cause of brown howler death in urban and suburban populations in southern Brazil [23, 57] explains the highly

cautious behavior and vigilance displayed by brown howlers when descending to the ground to drink, a behavior also observed in other primates (e.g. *Callithrix flaviceps* [11]). This threat is believed to reduce (or even eliminate) howlers' use of terrestrial water reservoirs in better-conserved large forests inhabited by wild carnivore populations in Central America (e.g. *A. palliata* [30, 58]). The frequency of brown howler remains in ocelot (*Leopardus pardalis*) scats in a ca. 950-ha Atlantic Forest reserve in southeastern Brazil highlights their vulnerability to wild felids [21].

The lower drinking in the spring and the overall positive influence of temperature on drinking behavior are likely explained, at least partially, by three main reasons. First, unlike at lower tropical latitudes where the hottest and driest times often coincide (i.e. dry season [59]), summer and spring are the hottest, but not the driest seasons in this subtropical study region (ca. 31˚S, S1 Fig in S1 File). In fact, rainfall is relatively well distributed throughout the year in Rio Grande do Sul state ([39], see also S1 and S2 Figs in S1 File), where 'rainy quarters' occur at any time [60]. Second, the higher availability and consumption of flowers and other water-richer/lower-secondary metabolite food items, such as young leaves [61] and some brown howler's preferred fruits (e.g. *Ficus* spp. and *Allophylus edulis*), and the consequent lower consumption of mature leaves, by the study groups also occurred in the hottest seasons [37]. This diet composition likely reduces the need for water to detoxify secondary metabolites in leafy material (hypothesis not supported in this study) while supplying water to counterbalance the losses of thermoregulation and many other vertebrate physiological processes (see [2, 4]). Therefore, the existence of a strong negative relationship between flower consumption and drinking behavior in brown howlers is not surprising. Flowers exploited by primates can have high water contents [62] that contribute to satisfy the animals' daily requirements as has been reported in some Neotropical primates (e.g. *Cebus imitator* [63]).

Finally, brown howlers may lower the thermoregulatory demands for water by preventing body over-heating and dehydration via positional adjustments and shade-seeking [64] during the hottest times of the day (strategies also reported in other Neotropical primates: *A. palliata* [65], *A. caraya* [36], *C. imitator* [66], *Callicebus bernhardi* [67]). Despite these strategies, the peaks of drinking in the afternoon tended to occur around the warmer times of the day (S5 Fig in S1 File), which are likely triggered by the need of water in this period of intensified physiological thermoregulation together with the recovery of the water spent earlier in the day (particularly during early and middle morning, when brown howlers are more active: OMC and GPH, personal observation) that is required to reach the homeostasis of blood osmolarity [34, 35]. Therefore, we found support for the TDH hypothesis. Future studies noninvasively assessing individuals' body temperature and water balance together with behavioral and climatological data are required to confirm that the drinking behavior of brown howlers is better explained by the thermoregulatory/dehydration-avoidance hypothesis than the metabolite detoxification hypothesis.

In sum, we found that the drinking behavior of brown howlers responded to changes in the consumption of flowers, rainfall and the thermal environment. Extrapolating from brown howlers to arboreal folivorous-frugivorous mammals in general that also lack adaptations to tolerate high levels of dehydration, we suggest that the higher the ambient temperature and lower the availability of water-rich plant items, the greater might be the challenges in fulfilling their water requirements, particularly in habitats where terrestrial water reservoirs are scarce, strongly seasonal or absent, such as small and medium forest fragments. Despite the higher availability of leaves than flowers and fruits in forests, highly folivorous mammals may also be more vulnerable to predators if they are forced to descend to the ground to drink from terrestrial reservoirs, particularly in forest fragments immersed in anthropogenic landscapes (a growing scenario in the tropics [68]), where dogs roam freely. In this respect, studies assessing

how differences in land-use and human disturbance influence the abundance and distribution of arboreal and terrestrial water reservoirs and how they impact the drinking behavior, water balance, and health of arboreal folivorous-frugivorous mammals are critical for enabling us to design and implement appropriate management strategies for promoting their conservation in anthropogenic fragmented landscapes.

## Supporting information

**S1 File.**
(DOCX)

## Acknowledgments

We thank Danielle Camaratta and João Claudio Godoy for logistical support and field assistance. We thank the landowners of the study fragments in Porto Alegre and Viamão for giving us permission to conduct this research on their properties. We thank Commandant Aluísio S. R. Filho for giving us permission to work in the Campo de Instrução de Santa Maria (CISM).

## Author Contributions

**Conceptualization:** Óscar M. Chaves, Júlio César Bicca-Marques.

**Data curation:** Óscar M. Chaves, Vanessa B. Fortes, Gabriela P. Hass.

**Formal analysis:** Óscar M. Chaves, Júlio César Bicca-Marques.

**Funding acquisition:** Kathryn E. Stoner, Júlio César Bicca-Marques.

**Investigation:** Óscar M. Chaves, Vanessa B. Fortes, Gabriela P. Hass, Renata B. Azevedo, Júlio César Bicca-Marques.

**Methodology:** Óscar M. Chaves, Vanessa B. Fortes, Gabriela P. Hass, Kathryn E. Stoner, Júlio César Bicca-Marques.

**Project administration:** Óscar M. Chaves.

**Supervision:** Óscar M. Chaves.

**Visualization:** Óscar M. Chaves, Júlio César Bicca-Marques.

**Writing – original draft:** Óscar M. Chaves.

**Writing – review & editing:** Óscar M. Chaves, Vanessa B. Fortes, Gabriela P. Hass, Renata B. Azevedo, Kathryn E. Stoner, Júlio César Bicca-Marques.

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
