## [Decision Letter · Decision Letter 0]

19 Nov 2020

PONE-D-20-21970

Leaf and flower consumption modulate the drinking behavior in a folivorous-frugivorous arboreal mammal

PLOS ONE

Dear Dr. Chaves,

Thank you for submitting your manuscript to PLOS ONE. After careful consideration, we feel that it has merit but does not fully meet PLOS ONE’s publication criteria as it currently stands. Therefore, we invite you to submit a revised version of the manuscript that addresses the points raised during the review process.

I agree with the reviewer's recommendation of a major revision. 

We look forward to receiving your revised manuscript.

Kind regards,

Julie Jeannette Gros-Louis, PhD

Academic Editor

PLOS ONE

Journal Requirements:

Reviewers' comments:

Reviewer's Responses to Questions

**Comments to the Author**

1. Is the manuscript technically sound, and do the data support the conclusions?

Reviewer #1: Yes

Reviewer #2: Yes

2. Has the statistical analysis been performed appropriately and rigorously? 

Reviewer #1: Yes

Reviewer #2: N/A

3. Have the authors made all data underlying the findings in their manuscript fully available?

Reviewer #1: Yes

Reviewer #2: Yes

4. Is the manuscript presented in an intelligible fashion and written in standard English?

Reviewer #1: Yes

Reviewer #2: Yes

5. Review Comments to the Author

Reviewer #1: This manuscript presents data on water ingestion by a tropical primate, and compares the demonstrated drinking patterns to two hypotheses that explain variation in the urge for animals to drink. The manuscript is relatively straightforward and contains valuable information. Most research on drinking in primates is sparse and anecdotal, so this thorough and long-term report will be a welcome addition to the literature. I have a few suggestions and considerations for the authors that I believe may benefit the paper.

First, for the GLMM analysis (l. 233), I recommend that the authors standardize the number of drinking events by group size. The dependent variable in this analysis is the number of daily drinking records. However, these data come from 14 howler groups, with differing sizes ranging from 6-12 animals. It seems logical to me that a group of 12 animals will have more observed drinking events than a group half that size. Dividing the daily drinking records by group size would be an easy way to fix this confounding factor without adding another predictor variable to the analysis. While group ID is a factor in the model, this nominal category will not control for effects of increasing or decreasing size between groups.

I think that more details and clarity on the relationship between fruit and leaf (and to a lesser extent, flower) consumption would be helpful. The methods indicate that leaf, fruit, and flower consumption are not multicollinear (l. 241-242). However, this is quite surprising for howlers, whose diet is essentially 100% comprised of these three foods. Are the howlers in this study eating some other items that decrease the correlation between these dietary items? For howlers, fruit and leaf consumption are generally inversely related to one another. If the percentage of leaf, flower, and fruit eaten in a day totals ~100%, then shouldn’t these predictor variables be correlated? Presenting a correlation matrix of these variables for readers may be helpful.

Furthermore, this relationship has important implications for the results. The authors find a significant effect of leaf consumption, but not fruit consumption. If fruit and leaf consumption are correlated, then this may simply be a statistical artifact rather than a biological pattern—leaf consumption is already explaining the variation in drinking records, so no variance is attributed to fruit. This is how multicollinear variables generally behave. Was a step function used so that leaf consumption was always entered into the model first? Again, knowing the correlation matrix between these dietary variables would help the authors establish this as a relevant biological finding vs. a statistic effect of correlated predictor variables.

Lastly, the interpretation of these data seems to blur the lines of this relationship. L. 342-346 uses fruit (and flower) consumption to explain the significant reduction in leaf consumption, which is interpreted as both increasing the amount of water ingested and reducing the need for water to detoxify secondary metabolites. In short, fruit are the most water rich food being consumed, which in turn drives lower leaf consumption, which then drives drinking behavior….despite that fruit was not a significant predictor of drinking behavior. Clearly, this relationship is family complex, however the conclusion that fruit plays a role in the observed drinking patterns, despite the results indicating otherwise, seems somewhat circular and/or inconsistent. Overall, a more detailed portrayal of how these variables are interrelated would help readers better understand these patterns.

Line by line comments

l. 3 Recommend deleting “the”

l. 33-34 Could some of this variation be due to group size rather than true drinking differences?

l. 47 ‘preformed’ is not defined until later in the ms. This may be unclear for abstract readers.

l. 79 I think the double use of ‘foods’ here is a bit awkward.

l. 133-135 A bit more information about the time lag of this process would be helpful. If hotter temperatures are experienced in the afternoon, would the drinking response be that quick to initiate?

l. 179 Is there a possibility for differential visibility between fragments? This could impact observed drinking.

l. 204-206 How are these feeding data summarized into the GLMM data? It looks like the GLMM had daily summaries...was the food item per each bout (and if so how were bouts defined)? Per minute? Was there ever averaging of a day's eating across more than one focal animal? Data were also collected via both instantaneous scan and focal animal sampling (l. 198-199)...how were feeding data from these different collection methods merged? Or were feeding data only from one of these methods?

l. 206-207 Does ‘individual records’ refer to individual drinks or individual monkeys?

l. 211-219 Temperature data were collected two different ways. Unless I am mistaken, the ms does not clarify which temperature measurements were used for which analyses.

l. 222-223 It would help to clarify in this text that a chi square test was done separately for each group.

l. 233-236 As per above, I recommend standardizing group drink records by the number of group members. Also, which ambient temperature measurement was used here?

l. 241-242 As above, it is surprising that VIF<3 if leaf, fruit, and flower consumption cumulatively make up ~100% of howler diet….shouldn’t there be a strong correlation? A correlation matrix would be helpful. This could be included at supplemental material, if needed.

l. 262-263 What is an 'event of group drinking' compared to 'individual drinking records?' The methods only defines "all drinking events."

l. 281-282 Same here—a ‘single drinking event’ is unclear. Is this the same as ‘individual drinking record’ earlier? Where is the ‘out of five’ coming from…particularly as some groups had >5 members?

l. 285 It would be helpful for readers to define the analyzed seasons earlier in the methods.

l. 289-291 I am not entirely sure what this sentence refers to. Is it that one, two, or three seasons were significantly higher than the remaining season(s)?

l. 293 Recommend specifying that pooling is of fragments.

l. 302 It is slightly unclear what ‘all’ refers to. I think this means that all predictor variables are cumulatively covered across the six different models, but this sentence could be confusing for readers.

l. 308 I don’t think ‘direct’ is the best word here. A direct relationship could still be either negative or positive.

l. 309 Recommend ‘nonsignificant’

l. 315 A comparison somewhere to overall activity patterns by time of day would be helpful. Are howlers simply more active in the afternoon?

l. 337-341 This would be helpful to know earlier. Recommend moving to the methods.

l. 342-346 As per above, the reasoning here seems slightly circular/inconsistent. Fruit consumption drives leaf consumption, which drives drinking behavior, but fruit consumption is not a significant predictor of drinking? Despite it being the most water-rich food ingested?

l. 348 It is unclear how this is a pers obs. How do you observe dehydration and overheating? I recommend deleting or being more specific about the observed behavior.

l. 351-355 Information on biological lags in the mammalian thirst response would be helpful, if known. Information on the overall timing of activities across the day would help as well.

l. 357 It is unclear what data this refers to...is this from the time of day analysis? I don’t see a direct analysis of temperature and within-day patterns. If so, recommend using the exact variables here so that readers can refer back to the results more easily.

l. 359 It is unclear what ‘broader temporal scale’ refers to. The GLMM was based on daily drinking…is that what this means?

Table 2: Recommend providing exact p values.

l. 603 The stated number does not appear in the middle of each bar, only a letter.

l. 612 (c) and (d) are not defined here. Also, this (a) and (b) may be confusing with those in l. 602-603.

Reviewer #2: A very nice study on the drinking patterns of a primate species, the brown howler monkeys. The study is based on an impressive sample size, in terms of study sites (fragments), individuals, and observation effort. Authors characterize the drinking sources, seasonal variation in drinking, and model drinking in relation to several dietary and environmental factors. The manuscript is well framed in the introduction; methods are generally complete and clearly described (but see specific comments); results are well-presented and statistical reporting is exhaustive (supplemental results are also very useful); and the discussion is synthetic and stimulates further research. The manuscript is also well organized and easy to read. I believe that this is an important contribution to all of those interested in drinking behavior. I only have a few comments detailed below. I especially call your attention to my questions concerning data organization and statistical analysis.

L131-133: as stated, it is not clear why you have two different predictions pertaining to the putative influence of diet composition on drinking behavior.

L181-182: I believe that habituation refers to individuals, not groups. How did you habituate individuals?

L199: which recording technique was used with the focal animal sampling?

L205: mature leaves?

L207: given that two different sampling methods were used to collect dietary data, how was “number of feeding records” calculated?

L214: “better” than what?

L217: meaning of “after each behavioral sampling unit”?

L223: you should describe before (behavioral data collection?) which water sources are you referring to.

L225: I apologize, but it is unclear to me how were these “total number of records” calculated given that two different sampling methods were used.

L225-226: the same argument could be used against the model you describe in the next paragraph so, am I missing something? Additionally, isn’t the calculation of proportions an actual way to account for such variation in sampling effort? Finally, please acknowledge that Figure 1 depicts data per group and per season, which you state here that could/should not be done.

L233: a single model was built (the complete model).

L234: about “contribution of fruits, leaves, and flowers to the diet”, how were these data organized? Which feeding records did you use to match drinking events? For instance, if in a particular day a single drinking event was recorded and it happened before noon, did you use only morning records as predictor values? If so, how was variation in the number of feeding records associated with drinking events accounted for? If not, please offer more information on data handling.

L234: perhaps add “(categorical variable with, three levels)” after “fragment size” to help us remembering your classification of fragment sizes.

Please report results of overdispersion diagnosis.

L307-308 & Table 2: the information theoretic framework is not based on significance testing. I refer you to Burnham and Anderson (2002), which you cite and, for instance, Mundry (2011; https://doi.org/10.1007/s00265-010-1040-y). The combination of “frequentist” and IT approaches is associated with Type I errors and is incorrect. You must decide which approach best suits your study. SE and CI shown in Table 2 are sufficient to assess the reliability of effects of predictors on drinking behavior.

Merge paragraphs starting at L336, L342 & L347.

6. PLOS authors have the option to publish the peer review history of their article (what does this mean?). If published, this will include your full peer review and any attached files.

Reviewer #1: No

Reviewer #2: **Yes: **Pedro Dias

---

## [Author Response · Author response to Decision Letter 0]

15 Jan 2021

*The responses to the reviewers and Editor are included in the Cover Letter. We also included the same responses below:

RESPONSES TO COMMENTS OF THE EDITOR AND REVIEWERS

Below we list our answers (italics) after the comments of the academic editor and the reviewer (bold)

Leaf and flower consumption modulate the drinking behavior in a folivorous-frugivorous arboreal mammal

PLOS ONE

Dear Dr. Chaves,

Thank you for submitting your manuscript to PLOS ONE. After careful consideration, we feel that it has merit but does not fully meet PLOS ONE’s publication criteria as it currently stands. Therefore, we invite you to submit a revised version of the manuscript that addresses the points raised during the review process.

I agree with the reviewer's recommendation of a major revision. 

We look forward to receiving your revised manuscript.

Kind regards,

Julie Jeannette Gros-Louis, PhD

Academic Editor

PLOS ONE

Journal Requirements:

Authors: Ok, thank you Julie. We have checked all technical details you have mentioned. Below we have included detailed responses to the comments and suggestions of the two reviewers.

Reviewers' comments:

Reviewer's Responses to Questions

Comments to the Author

1. Is the manuscript technically sound, and do the data support the conclusions?

Reviewer #1: Yes

Reviewer #2: Yes

2. Has the statistical analysis been performed appropriately and rigorously?

Reviewer #1: Yes

Reviewer #2: N/A

3. Have the authors made all data underlying the findings in their manuscript fully available?

Reviewer #1: Yes

Reviewer #2: Yes

4. Is the manuscript presented in an intelligible fashion and written in standard English?

Reviewer #1: Yes

Reviewer #2: Yes

5. Review Comments to the Author

Reviewer #1: 

This manuscript presents data on water ingestion by a tropical primate, and compares the demonstrated drinking patterns to two hypotheses that explain variation in the urge for animals to drink. The manuscript is relatively straightforward and contains valuable information. Most research on drinking in primates is sparse and anecdotal, so this thorough and long-term report will be a welcome addition to the literature. I have a few suggestions and considerations for the authors that I believe may benefit the paper.

Authors: Thank you for the comments.

First, for the GLMM analysis (l. 233), I recommend that the authors standardize the number of drinking events by group size. The dependent variable in this analysis is the number of daily drinking records. However, these data come from 14 howler groups, with differing sizes ranging from 6-12 animals. It seems logical to me that a group of 12 animals will have more observed drinking events than a group half that size. Dividing the daily drinking records by group size would be an easy way to fix this confounding factor without adding another predictor variable to the analysis. While group ID is a factor in the model, this nominal category will not control for effects of increasing or decreasing size between groups.

Authors: We understand the concern of the reviewer. However, we believe that this standardization is not necessary. Despite the differences in group size, there was no correlation between group size and the number of drinking records per day (rs=0.03, P=0.54) probably, at least partially, because not all group members used to take part of the drinking sessions/events as we have mentioned in Results (p.13, line 306). Furthermore, the proposed standardization would transform the count data into proportions and, then, the transformed data would not fit the Poisson (nor negative binomial distribution or Gaussian even after several transformations) distribution family.

Furthermore, we have included group size as a covariable in the GLMM model to control for any potential influence of differences in this variable on the response variable. We have clarified this point in the Methods (please see p.11, lines 253-255).

I think that more details and clarity on the relationship between fruit and leaf (and to a lesser extent, flower) consumption would be helpful. The methods indicate that leaf, fruit, and flower consumption are not multicollinear (l. 241-242). However, this is quite surprising for howlers, whose diet is essentially 100% comprised of these three foods. Are the howlers in this study eating some other items that decrease the correlation between these dietary items? For howlers, fruit and leaf consumption are generally inversely related to one another. If the percentage of leaf, flower, and fruit eaten in a day totals ~100%, then shouldn’t these predictor variables be correlated? Presenting a correlation matrix of these variables for readers may be helpful.

Authors: We agree that the lack of multicollinearity between the consumption of leaves, fruits and flowers could sound counterintuitive. However, as we disclosed in the Methods, we analyzed “the number of feeding records devoted to each plant item” instead of the percentage of records devoted to them. The collinearity would be expected in the latter case, not necessarily in the former because the howlers can simultaneously increase or decrease the number of feeding records with both leaves and fruits, for example. This difference explains why we found a negligible collinearity between these variables using the Variation Inflation Factor (p.12, line 266). We have cited a graph with the correlation matrix between the dietary variables (Fig S3 in the Supplementary Material) in the Methods (p.12, line 267). Overall, the correlation coefficients between the consumption of fruits, leaves and flowers expressed by the number of feeding records ranged from 0.01 to -0.21. 

Furthermore, this relationship has important implications for the results. The authors find a significant effect of leaf consumption, but not fruit consumption. If fruit and leaf consumption are correlated, then this may simply be a statistical artifact rather than a biological pattern—leaf consumption is already explaining the variation in drinking records, so no variance is attributed to fruit. This is how multicollinear variables generally behave. 

Authors: We are confident the pattern we report is not a statistical artifact. Please see our explanation in our response above.

Was a step function used so that leaf consumption was always entered into the model first? Again, knowing the correlation matrix between these dietary variables would help the authors establish this as a relevant biological finding vs. a statistic effect of correlated predictor variables.

Authors: Yes, we used a step function and the correlation matrix support our findings as can be seen in the correlation matrix in the Supplementary Fig S3:

Fig. S3. Matrix of Spearman and Pearson (in parentheses) correlation coefficients between the consumption of fruits, leaves and flowers by14 brown howler monkey (Alouatta guariba clamitans) groups in the State of Rio Grande do Sul, Brazil.

Lastly, the interpretation of these data seems to blur the lines of this relationship. L. 342-346 uses fruit (and flower) consumption to explain the significant reduction in leaf consumption, which is interpreted as both increasing the amount of water ingested and reducing the need for water to detoxify secondary metabolites. In short, fruit are the most water rich food being consumed, which in turn drives lower leaf consumption, which then drives drinking behavior….despite that fruit was not a significant predictor of drinking behavior. Clearly, this relationship is family complex, however the conclusion that fruit plays a role in the observed drinking patterns, despite the results indicating otherwise, seems somewhat circular and/or inconsistent. Overall, a more detailed portrayal of how these variables are interrelated would help readers better understand these patterns.

Authors: We agree that this paragraph may sound circular. However, we are not referring here to all fruits, but particularly to some preferred fleshy fruit species abundant in this season and to other water-richer items other than adult leaves, such as young leaves. It is reasonable to expect that the consumption of all these water-richer plant items together contributes to reduce the need of drinking. We rewrote the paragraph in an attempt of making our rationale clearer. Please see p. 16, lines 370-372.

Line by line comments

l. 3 Recommend deleting “the”

Authors: Done.

l. 33-34 Could some of this variation be due to group size rather than true drinking differences?

Authors: No, we do not believe so. As we explained above (p. 5), there was no correlation between group size and the number of drinking records. 

We suspect that the differences were partially influenced by the availability of water sources (i.e. streams, rivers, tree holes, and bromeliads) in each study site, in addition to the relationship with leaf and flower consumption. Unfortunately, we do not have reliable information on this issue, particularly on the availability of hard-to-detect water sources in the canopy, such as tree holes and bromeliad rosettes. 

l. 47 ‘preformed’ is not defined until later in the ms. This may be unclear for abstract readers.

l. 79 I think the double use of ‘foods’ here is a bit awkward.

Authors: The text was corrected.

l. 133-135 A bit more information about the time lag of this process would be helpful. If hotter temperatures are experienced in the afternoon, would the drinking response be that quick to initiate?

Authors: In fact, the temperature in the study sites increased gradually throughout the day, reaching a peak often later in the afternoon (please see Fig. S4). Because of this, we predicted a similar pattern in the drinking. We have edited the text in this section for clarity. 

l. 179 Is there a possibility for differential visibility between fragments? This could impact observed drinking.

Authors: Despite differences (not measured, but probably subtle) in visibility both between sites as well as within each site, we do not believe this biased our ability to record events of drinking across studies. The canopy of all Atlantic Forest study sites was sufficiently open to enable us to monitor most group members throughout the day. We have clarified this issue in the Methods (see p. 7, lines 160-161).

l. 204-206 How are these feeding data summarized into the GLMM data? It looks like the GLMM had daily summaries...was the food item per each bout (and if so how were bouts defined)? Per minute? 

Was there ever averaging of a day's eating across more than one focal animal? Data were also collected via both instantaneous scan and focal animal sampling (l. 198-199)...how were feeding data from these different collection methods merged? Or were feeding data only from one of these methods?

Authors: We have improved the description of the methods used to record diet and drinking behavior data (please see p. 9). The data on feeding behavior were recorded via instantaneous scan sampling at 15-min intervals in most study periods. The exception was the period (v) of groups S4, S5 and S6, whose data were collected via 5-min focal samples at 10-min intervals. We recorded information on the plant species exploited, the food item (s) consumed, and the time devoted to each species/item per study group (scan) or the focal monkey (focal-animal). 

We organized the data on diet composition and drinking behavior based on the number of records per study day with at least one drinking record to run the analyses (available in Mendeley Data: http://dx.doi.org/10.17632/3gxy6vrsbf.1).

l. 206-207 Does ‘individual records’ refer to individual drinks or individual monkeys?

Authors: We refer here to individual drinking events or drinking records, not to individual monkeys. We removed the word ‘individual’ to avoid confusion.

l. 211-219 Temperature data were collected two different ways. Unless I am mistaken, the ms does not clarify which temperature measurements were used for which analyses.

Authors: The temperature data that we used to run the GLMM were obtained from the same source: Instituto Nacional de Meteorologia do Brasil. However, we used hourly in situ variations in temperature to prepare Fig S5. We clarified this point in the text (please see p. 10).

l. 222-223 It would help to clarify in this text that a chi square test was done separately for each group.

Authors: Done (see p. 10, line 239)

l. 233-236 As per above, I recommend standardizing group drink records by the number of group members. 

Authors: Please see our response on this issue above (p. 5).

Also, which ambient temperature measurement was used here?

Authors: As we mentioned above, we used the ambient temperature available from the Instituto de Meteorologia do Brasil (see p. 10, lines 223-224).

l. 241-242 As above, it is surprising that VIF<3 if leaf, fruit, and flower consumption cumulatively make up ~100% of howler diet….shouldn’t there be a strong correlation? A correlation matrix would be helpful. This could be included at supplemental material, if needed.

Authors: Please see our response on this issue in p. 5 above and the correlation matrix in Fig. S3.

l. 262-263 What is an 'event of group drinking' compared to 'individual drinking records?' The methods only defines "all drinking events."

Authors: We are referring to ‘drinking events’ (i.e. when one or more members of the study group drank). Corrected (please see p. 9, lines 199-200).

l. 281-282 Same here—a ‘single drinking event’ is unclear. Is this the same as ‘individual drinking record’ earlier? Where is the ‘out of five’ coming from…particularly as some groups had >5 members?

Authors: Corrected. Here we are referring to the number of monkeys participating in a ‘drinking event’, not to an ‘individual drinking record’. Then we observed that a drinking event involved between one-fifth (or 20%) to four-fifths (or 80%) of group members. Please see p. 13, line 306.

l. 285 It would be helpful for readers to define the analyzed seasons earlier in the methods.

Authors: Done. Please p. 10, line 241.

l. 289-291 I am not entirely sure what this sentence refers to. Is it that one, two, or three seasons were significantly higher than the remaining season(s)?

Authors: Corrected. See p. 14, lines 314-317

l. 293 Recommend specifying that pooling is of fragments.

Authors: The ‘pooled dataset’ refers to the comparison of the number records in each season considering all groups together. We improved the sentence for clarity (please see p. 14, lines 317-318).

l. 302 It is slightly unclear what ‘all’ refers to. I think this means that all predictor variables are cumulatively covered across the six different models, but this sentence could be confusing for readers.

Authors: We are referring to the six study predictor variables mentioned in Methods. In fact, now there are eight variables after the addition of ‘group size’ and ‘sampling method’ as covariables to address the recommendations of reviewer #2 (please see p. 11). We found nine models with ΔAICc<2 in the new GLMM analyses (please see the new Table 2). We edited the sentence to avoid confusion (please see p. 14, line 328).

l. 308 I don’t think ‘direct’ is the best word here. A direct relationship could still be either negative or positive.

Authors: We believe the word ‘direct’ is not statistically incorrect, particularly because we previously mention the word ‘inverse’. However, we rewrote this sentence following reviewer 2’s suggestion about the incorrect use of the Null Hypotheses Significance Testing NHST combined with IT-models (please see p. 15 below and the manuscript’s p. 15).

l. 309 Recommend ‘nonsignificant’

Authors: We removed this term and any reference to p-values in this paragraph in light of reviewer 2’s recommendation about the Null Hypothesis Testing approach (see p. 15).

l. 315 A comparison somewhere to overall activity patterns by time of day would be helpful. Are howlers simply more active in the afternoon?

Authors: We have no data on the activity budget of all 14 study groups and this analysis was not an objective of this manuscript (it will be part of another manuscript with a subset of six study groups). Nevertheless, we have the impression based on our field observations that brown howlers are more active early in the morning than in the afternoon. They often spend more time feeding in the morning (particularly from 6:00 to 11:00) and resting in the shade from 11:30 to 14:00-15:00 (depending on ambient temperature). However, some individuals occasionally drank from arboreal or terrestrial water sources during the afternoon resting periods. We added this information in the new version (please see p. 17, lines 387-388).

l. 337-341 This would be helpful to know earlier. Recommend moving to the methods.

Authors: We disagree with this recommendation because this sentence is a potential explanation for our finding, not a simple methodological detail. Furthermore, we provided a detailed description of the climatic conditions of the study sites in Methods (p. 7), where we highlighted that there is not a clear rainfall difference between months.

l. 342-346 As per above, the reasoning here seems slightly circular/inconsistent. Fruit consumption drives leaf consumption, which drives drinking behavior, but fruit consumption is not a significant predictor of drinking? Despite it being the most water-rich food ingested?

Authors: As we explained above (p. 6), we improved this sentence to avoid this circular argument (please also see the corrected sentence in p. 16, lines 370-376).

l. 348 It is unclear how this is a pers obs. How do you observe dehydration and overheating? I recommend deleting or being more specific about the observed behavior.

Authors: Here we are referring to a frequent behavioral pattern observed in some study groups (specifically groups S2, S3, S7, M2, L1, L4, and L5): these groups used to select a riparian-shaded site to rest during the hottest hours of the day (partially midday). We considered it an important thermoregulatory behavior because according to our meteorological field records for six of these groups (S1, S2, S3, L1, L2, and L3), the temperature in the riparian forest can be up to 3ºC lower than in forest edges and other open forested areas. The other groups did not show this behavior because their habitats had no riparian forest.

However, we removed this sentence from the text (see p. 17, lines 381-382) because we do not have data on thermoregulation or dehydration as highlighted by the reviewer and because this idea is also implicit in the ‘shade-seeking adjustments’ mentioned in the next sentence.

l. 351-355 Information on biological lags in the mammalian thirst response would be helpful, if known. Information on the overall timing of activities across the day would help as well.

Authors: Unfortunately, we did not find information on biological lags in the mammalian thirst response in our literature review. However, as we mentioned above (p. 9), we improved the sentence by including additional information on the activity periods of brown howlers (please see p. 17, lines 387-388).

l. 357 It is unclear what data this refers to...is this from the time of day analysis? I don’t see a direct analysis of temperature and within-day patterns. If so, recommend using the exact variables here so that readers can refer back to the results more easily.

Authors: Yes, here we are referring to the fact that in some groups the peaks of drinking records (Fig 3) trended to occur in the afternoon (Fig S5). We further explain that we did perform a specific analysis on this issue. We rewrote the paragraph to avoid further confusions (please see p. 17).

l. 359 It is unclear what ‘broader temporal scale’ refers to. The GLMM was based on daily drinking…is that what this means?

Authors: As mentioned in the previous response, this sentence was improved to avoid confusion. 

Table 2: Recommend providing exact p values.

Authors: We removed the P-values in light of reviewer 2’s recommendations (please see p. 15 below). We followed this recommendation because many statisticians have shown that the combination of IT-based inference and Null Hypotheses Significance Testing NHST is not statistically appropriate given that it significantly increases Type I error. In fact, as reviewer 2 mentioned, the C.I. and the S.E. provide sufficient evidence to assess the relevance of predictor variables. Please see Mundry (2011, https://doi.org/10.1007/s00265-010-1040-y) for an excellent explanation on this issue.

l. 603 The stated number does not appear in the middle of each bar, only a letter.

Authors: Corrected.

l. 612 (c) and (d) are not defined here. Also, this (a) and (b) may be confusing with those in l. 602-603.

Authors: Here we are referring to Figs 2a and 2b, not to the letters into the bars. We edited the Fig. 2. legend to avoid confusing readers.

Reviewer #2: 

A very nice study on the drinking patterns of a primate species, the brown howler monkeys. The study is based on an impressive sample size, in terms of study sites (fragments), individuals, and observation effort. Authors characterize the drinking sources, seasonal variation in drinking, and model drinking in relation to several dietary and environmental factors. The manuscript is well framed in the introduction; methods are generally complete and clearly described (but see specific comments); results are well-presented and statistical reporting is exhaustive (supplemental results are also very useful); and the discussion is synthetic and stimulates further research. The manuscript is also well organized and easy to read. I believe that this is an important contribution to all of those interested in drinking behavior. I only have a few comments detailed below. I especially call your attention to my questions concerning data organization and statistical analysis.

Authors: Thank you very much for your evaluation and your well-justified comments and suggestions. In this revised version of the manuscript we have corrected most of the issues mentioned by you and the other reviewer.

L131-133: as stated, it is not clear why you have two different predictions pertaining to the putative influence of diet composition on drinking behavior.

Authors: We did not understand why the reviewer considered that we have two predictions about the influence of diet composition… In this first prediction we simply expect that howlers will complement the preformed water in the diet with drinking water as a response to the highly seasonal pattern of availability of water-rich food items in the region. This prediction is based on previous research that found that howlers often supplement the preformed water in their diet with water from arboreal or terrestrial sources. We found support for this prediction as you can see in Fig. 2. In the second prediction we refer to the within-day gradual increase in drinking. Then, our third prediction addresses the main drivers of drinking behavior. In sum, we believe that our hypotheses are not overlapping and complement each other to improve our understanding of the factors affecting the drinking behavior of howler monkeys.

L181-182: I believe that habituation refers to individuals, not groups. How did you habituate individuals?

Authors: Yes, we refer to the habituation of the individuals of each study group. To habituate the howlers in medium and large fragments we followed most recommendations of Williamson & Feistner (2011). We improved the paragraph by including additional details on the habituation process (please see p. 8, lines 174-182).

Williamson EA, Feistner ATC. Habituating primates: processes, techniques, variables and ethics. In: Setchell JM, Curtis DJ, editors. Field and laboratory methods in primatology: a practical guide. New York: Cambridge University Press, 2011. pp. 33–50.

L199: which recording technique was used with the focal animal sampling?

Authors: We applied the focal-animal method in a single research – period (v). The method consisted in recording the behavior, plant species exploited and food item(s) ingested by the focal howler instantaneously at 20-s intervals during 5-min focal sampling units every 15 min. A single focal adult was monitored per group during each study period in order to fulfill the major goals of the respective research. We identified individuals according to their sex-age category, and based on their size, pelage color, skin pigmentation, face form, genital’s size and color, scars, etc. We added some of these details in the revised text (please see p. 9).

L205: mature leaves?

Authors: Yes, we collected information on mature (i.e. old or adult) and young leaves. We changed ‘old’ to ‘adult’ (as is frequently used by plant physiologists) to avoid confusion.

L207: given that two different sampling methods were used to collect dietary data, how was “number of feeding records” calculated?

Authors: We clarified these methodological details in the Methods section (please see p. 9, lines 207-217). The number of feeding records on each plant item by the focal animal was the sum of the instantaneous records (once every 20 s during the 5-min focal sampling units), whereas the number of instantaneous scan sampling feeding records was the sum of all records of feeding on a given food item by all recorded group members.

L214: “better” than what?

Authors: Here we are explaining that we used the mean ambient temperature and weekly rainfall as proxies or indicators of thermal ambient temperature and water availability for brown howlers…To avoid confusion, we have replaced the words ‘they represent better’ by ‘proxies of’ (see p. 10, line 224-227).

L217: meaning of “after each behavioral sampling unit”?

Authors: Corrected. We replaced ‘behavioral sampling unit’ with ‘scan or focal sampling unit’.

L223: you should describe before (behavioral data collection?) which water sources are you referring to.

Authors: Done (please see p. 9, line 215).

L225: I apologize, but it is unclear to me how were these “total number of records” calculated given that two different sampling methods were used.

Authors: Please see our previous comment on this issue (L207) above. Furthermore, we have clarified these details in the Methods (p. 9).

L225-226: the same argument could be used against the model you describe in the next paragraph so, am I missing something? Additionally, isn’t the calculation of proportions an actual way to account for such variation in sampling effort? 

Authors: We feel that the reviewer misinterpreted the meaning of this sentence. In fact, our point is that the inter-fragment comparisons may be inappropriate because of differences in sampling effort (see Table 1). For this reason, we only compared the proportion of records devoted to each water source or season within each study fragment. We tried to clarify this issue in the Methods of the new version (please see p. 11).

Finally, please acknowledge that Figure 1 depicts data per group and per season, which you state here that could/should not be done.

Authors: We believe that the reviewer is referring to Figure 2, instead of Figure 1, which is the map. As we have explained in the previous comment, inter-fragment comparisons may be inadequate because of differences in sampling effort. For this reason we performed the comparisons shown in Figure 2 only within-fragments. We also have clarified this point in the Methods (p.11, line 242-243). 

L233: a single model was built (the complete model).

Authors: Corrected.

L234: about “contribution of fruits, leaves, and flowers to the diet”, how were these data organized? Which feeding records did you use to match drinking events? 

Authors: We used the information on the total number of drinking records, the weekly rainfall, the ambient temperature, and the number of feeding records devoted to fruits, leaves, and flowers (calculated as mentioned in the Methods, see p. 9) for each sampling day with at least one drinking record. Our datasets and all details of data organization are available in Mendeley Data (http://dx.doi.org/10.17632/3gxy6vrsbf.1).

For instance, if in a particular day a single drinking event was recorded and it happened before noon, did you use only morning records as predictor values? If so, how was variation in the number of feeding records associated with drinking events accounted for? If not, please offer more information on data handling.

Authors: No, we always used all feeding records of the day, irrespective if the hypothetic single drinking event happened before noon or near the end of the day. Therefore, our predictor variable was the total number of feeding records as we mentioned in the Methods (please see p. 11, line 252). 

L234: perhaps add “(categorical variable with, three levels)” after “fragment size” to help us remembering your classification of fragment sizes.

Authors: In fact, we used the total area of each study fragment, instead of size category, in this analysis as we mentioned in Table 1. We clarified this detail in the new version (please see p. 11, line 251).

Please report results of overdispersion diagnosis.

Authors: Done. We now report this parameter in the Methods. Despite the fact that our data are overdispersed (overdispersion ratio=1.4, P=0.03) as is often the case for most count data, we believe that the use of an alternative family distribution is not indispensable in our case (please see more details in p. 11, lines 257-260)

L307-308 & Table 2: the information theoretic framework is not based on significance testing. I refer you to Burnham and Anderson (2002), which you cite and, for instance, Mundry (2011; https://doi.org/10.1007/s00265-010-1040-y). The combination of “frequentist” and IT approaches is associated with Type I errors and is incorrect. You must decide which approach best suits your study. SE and CI shown in Table 2 are sufficient to assess the reliability of effects of predictors on drinking behavior.

Authors: This is a very good point! We acknowledge that the use of both statistical approaches (IT-based inference and NHST-Null Hypotheses Significance Testing) is not statistically appropriate despite its frequent use in the scientific literature as mentioned by Mundry (2011). We believe that the IT-based inference is the most appropriate approach for our data and concur that SE and CI (and the sum of the Akaike weights) are sufficient to assess the reliability and relevance of our predictor variables. Therefore, we improved this paragraph following the suggestion of both reviewers (please see p. 15). We also removed the z-values and the significance from Table 2 and the text. We only show the parameter values and the relative importance of the main predictors (∑wi) in the new version.

Merge paragraphs starting at L336, L342 & L347.

Authors: Done.

---

## [Editor Report · Decision Letter 1]

28 Jan 2021

Flower consumption, ambient temperature and rainfall modulate drinking behavior in a folivorous-frugivorous arboreal mammal

PONE-D-20-21970R1

Dear Dr. Chaves,

We’re pleased to inform you that your manuscript has been judged scientifically suitable for publication and will be formally accepted for publication once it meets all outstanding technical requirements.

Kind regards,

Julie Jeannette Gros-Louis, PhD

Academic Editor

PLOS ONE
---

## [Editor Report · Acceptance letter]

8 Feb 2021

PONE-D-20-21970R1 

Flower consumption, ambient temperature and rainfall modulate drinking behavior in a folivorous-frugivorous arboreal mammal 

Dear Dr. Chaves:

I'm pleased to inform you that your manuscript has been deemed suitable for publication in PLOS ONE. Congratulations! Your manuscript is now with our production department. 

Kind regards, 

on behalf of

Dr. Julie Jeannette Gros-Louis 

Academic Editor

PLOS ONE